# DEER-PREdict: Software for efficient calculation of spin-labeling EPR and NMR data from conformational ensembles

**Giulio Tesei**[1⦿], **João M. Martins**[1⦿], **Micha B. A. Kunze**[1], **Yong Wang**[1], **Ramon Crehuet**[1,2], **Kresten Lindorff-Larsen**[1]*

**1** Structural Biology and NMR Laboratory & the Linderstrøm-Lang Centre for Protein Science, Department of Biology, University of Copenhagen, Copenhagen, Denmark, **2** CSIC-Institute for Advanced Chemistry of Catalonia (IQAC), Barcelona, Spain

⦿ These authors contributed equally to this work.
* lindorff@bio.ku.dk

**Data Availability Statement:** The software is available on GitHub https://github.com/KULL-Centre/DEERpredict. DEER-PREdict is also distributed as a PyPI package (https://pypi.org/

## Abstract

Owing to their plasticity, intrinsically disordered and multidomain proteins require descriptions based on multiple conformations, thus calling for techniques and analysis tools that are capable of dealing with conformational ensembles rather than a single protein structure. Here, we introduce DEER-PREdict, a software program to predict Double Electron-Electron Resonance distance distributions as well as Paramagnetic Relaxation Enhancement rates from ensembles of protein conformations. DEER-PREdict uses an established rotamer library approach to describe the paramagnetic probes which are bound covalently to the protein.DEER-PREdict has been designed to operate efficiently on large conformational ensembles, such as those generated by molecular dynamics simulation, to facilitate the validation or refinement of molecular models as well as the interpretation of experimental data. The performance and accuracy of the software is demonstrated with experimentally characterized protein systems: HIV-1 protease, T4 Lysozyme and Acyl-CoA-binding protein. DEER-PREdict is open source (GPLv3) and available at github.com/KULL-Centre/DEERpredict and as a Python PyPI package pypi.org/project/DEERPREdict.

## Author summary

The accurate description of the structure of a protein is pivotal to fully understand its biological function. A large fraction of eukaryotic proteins is intrinsically disordered or consists of multiple folded domains connected by disordered regions. The structure of these proteins is highly flexible and can only be described by large ensembles of conformations. The characterization of these ensembles can be achieved by integrating *in silico* molecular modelling and simulations with experiments. Here, we present DEER-PREdict, an open-source software program to conveniently and efficiently calculate the observables of two biophysical methods, namely double electron-electron resonance (DEER) and paramagnetic relaxation enhancement (PRE) nuclear magnetic resonance. Both techniques provide distance information for highly dynamic systems and involve labelling proteins at

project/DEERPREdict) and archived on Zenodo (https://doi.org/10.5281/zenodo.3968394). DEER-PREdict is distributed under GPL license version 3.

**Funding:** M.B.A.K. acknowledges funding from the Lundbeck Foundation (lundbeckfonden.com). R.C. acknowledges funding from MINECO (CTQ2016-78636-P, https://www.mineco.gob.es/). K.L.-L. acknowledges funding via a Sapere Aude Starting Grant from the Danish Council for Independent Research (Natur og Univers, Det Frie Forskningsråd, 12-126214, https://dff.dk/) and the Lundbeck Foundation BRAINSTRUC initiative in structural biology (R155-2015-2666, lundbeckfonden.com). The funders had no role in study design, data collection and analysis, decision to publish, or preparation of the manuscript.

**Competing interests:** The authors have declared that no competing interests exist.

one or more sites with flexible probe molecules. The DEER-PREdict package combines previously developed and validated methods for placing multiple conformations of a nitroxide molecule at the protein sites with the rapid calculation of DEER and PRE observables from large ensembles of protein structures. Through examples, we illustrate the use of DEER-PREdict as a tool for interpreting experimental results, validating molecular models of flexible proteins as well as designing experiments.

This is a *PLOS Computational Biology* Software paper.

## Introduction

A detailed understanding of protein function often requires an accurate description of the structure and dynamics of a protein. The characterization of protein complexes as well as multi-domain and disordered proteins is typically achieved by combining experimental techniques of distinct spatial resolution [1]. Among the many different experimental techniques that may be used, we focus here on (i) a pulsed electron paramagnetic resonance (EPR) technique called double electron-electron resonance (DEER) and (ii) a nuclear magnetic resonance (NMR) method called paramagnetic relaxation enhancement (PRE). While the two methods differ substantially in their physics and applications, they have in common that they generally involve adding so-called spin-labels to the protein of interest.

DEER, also sometimes known as pulsed electron-electron double resonance (PELDOR), [2–6] relies on probing magnetic dipole-dipole interactions that are sensitive to distributions of residue-residue distances ranging from ∼1.8 nm to ∼8 nm, and up to 16 nm in deuterated soluble proteins [7–10]. For proteins, DEER generally requires site-directed spin labeling (SDSL) to functionalize a pair of selected residues with paramagnetic probes, e.g. 1-Oxyl-2,2,5,5-tetramethylpyrroline-3-methyl methanethiosulfonate (MTSSL) [4].

PRE NMR also makes use of SDSL to provide information on the average proximity of protein backbone nuclei up to ∼3.5 nm away from the unpaired electron of the paramagnetic probe [11]. The dependence of the rate of relaxation enhancement on the electron-proton distance, $r$, scales as $\langle r^{-6} \rangle$, making the measurement particularly sensitive to contributions from different probe conformations [11].

Since spin labels are conformationally dynamic, both protein and paramagnetic probes need to be described by conformational ensembles to obtain accurate predictions of DEER and PRE observables from molecular models [12–14]. Molecular dynamics (MD) simulations are one approach to obtain conformational ensembles that model the structure and dynamics of spin-labels for the calculation of EPR and NMR data [15–18]. While such analyses can provide unique insight into the motions of and interactions between protein and spin-label [19], they may be relatively expensive computationally. Further, many studies integrate results from multiple probe positions, or pairs thereof, which may be difficult to represent in a single MD simulation with explicit representations of the probes.

Another approach is to use conformational analysis of the spin-label combined with modelling of the dynamics [20–23]. Such analyses suggest that the conformational variation of spin-labelled sites is rotameric, i.e. it can be relatively well described by a finite number of defined structures. Thus, in the calculation of DEER data, rapid modeling of dynamic paramagnetic probes was made possible with the introduction of the rotamer library approach (RLA) applied to the MTSSL probe by Polyhach *et al.* [24].

Here, building and expanding on earlier work [3, 24–27], we developed a software tool for fast predictions of DEER and PRE observables from large conformational ensembles using the RLA. We present our implementation, distributed as the DEER-PREdict software, and test it against experimental data on HIV-1 Protease, T4 Lysozyme and the Acyl-CoA-Binding Protein. This software has been previously used for the calculation of both intra- and intermolecular DEER and PRE NMR data [28, 29], and has some overlap with the features in RotamerConvolveMD [25] (github.com/MDAnalysis/RotamerConvolveMD). DEER-PREdict is open-source, documented (deerpredict.readthedocs.io) and open to contributions from the community.

## Design and implementation

DEER-PREdict is written in Python and is available as a Python API, which facilitates its integration within larger data pipelines. Predictions of DEER and PRE data are carried out via the DEERpredict and PREpredict classes. Both classes are initialized with protein structures (provided as MDAnalysis [30] Universe objects) and spin-labeled positions (residue numbers and chain IDs). As shown in the *Results* section, the calculations are triggered by the *run* function, which also sets additional attributes such as the paths of input and output files as well as experiment-specific parameters. Per-frame data is saved in compressed binary files (HDF5 and pickle files) to allow for fast calculations of ensemble averages in reweighting schemes.

For the presented software, we adopt a procedure of rotamer placement and evaluation of labeled sites which is analogous to the RLA of Polyhach *et al.* [24], and we build on this previous work to implement fast calculations of DEER and PRE observables from large structural ensembles, such as MD trajectories.

## Rotamer library approach

Rotamer libraries have a long history in protein structural analysis [31], with an early application being to study side-chain packing [32]. Several other applications of this approach were later employed, e.g. in homology modeling and protein design [33, 34]. In our implementation, the RLA is used to insert the rotamer conformations of a paramagnetic probe at the spin-labeled site and to calculate the Boltzmann weight of each conformer. By default, we use the MTSSL 175 K rotamer library by Polyhach *et al.* [24], which was filtered to include only the $\chi_1$ $\chi_2$ conformations that are most commonly found in crystal structures of T4 Lysozyme [35]. As shown by Klose *et al.* [26], this selection criterion increases the accuracy of the calculated electron-electron distance distributions. The code is, however, general and it is possible to add new rotamer libraries by providing a text file containing the Boltzmann weights of each rotamer state $p_i^{int}$, a topology file (PDB format) and a trajectory file (DCD format) where rotamers are aligned with respect to the the plane defined by C$\alpha$ atom and C–N peptide bond. These files should be included in the *lib* folder and listed in the yaml file *DEERPREdict/lib/libraries.yml*. The default MTSSL 298 K MC/UFF C$\alpha$S$\delta$ rotamer libraries of the Matlab-based MMM modeling toolbox [13] are also provided in the DEER-PREdict package.

Following the alignment of the rotamer to the protein backbone (C$\alpha$, C and N atoms), the calculation of the Boltzmann weights is based on the sum of internal, $\epsilon_i^{int}$, and external, $\epsilon_i^{ext}$, energy contributions. The internal contribution is taken from Polyhach *et al.* [24] and results from the clustering of representative dihedral combinations from MD simulations. The normalized frequency of each cluster throughout the MD trajectory was used to determine the Boltzmann probability, $p_i^{int}$, of a given $i^{th}$ state, which readily can be converted into an internal energy contribution, $\epsilon_i^{int}$, via Boltzmann inversion. On the other hand, the external energy contribution is calculated on the fly as the dispersion interaction energy between heavy atoms of

rotamer and protein residues within a 1-nm cutoff, using the pairwise 6-12 Lennard-Jones potential of the CHARMM36 force field, with atom sizes scaled by the input parameter *sigma_scaling*, which defaults to 0.5 as in the MMM modeling toolbox (http://www.epr.ethz.ch/software) [13].

The overall probability of the $i^{th}$ rotamer state is then calculated as

$$p_i = p_i^{int} p_i^{ext} = p_i^{int} \frac{\exp\left(-\epsilon_i^{ext}/kT\right)}{Z} \tag{1}$$

where $Z = \sum_i p_i^{int} \exp\left(-\epsilon_i^{ext}/kT\right)$ is the steric partition function quantifying the fit of the rotamer in the embedding protein conformation. Low values of $Z$ result from large probe-protein van der Waals interaction energies, suggesting a tight placement of the spin label either due to a displacement of the rotamers or indicative of a wild-type conformation made inaccessible by the presence of the MTSSL probe. Especially in folded proteins, probes located in closely packed regions are likely to induce changes in the ensemble of the spin-labeled protein compared to the native form, and should be avoided in designing SDSL experiments. Therefore, in the calculation of DEER or PRE NMR observables, frames with $Z < 0.05$ are discarded to preclude spurious conformers from contributing to the ensemble average [24]. For the MTSSL 175 K rotamer library, a $Z$ cutoff of 0.05 is compatible with distributions of $\epsilon_i^{ext}$ values where at most one of the 46 rotamers has $\epsilon_i^{ext} \approx 3\ k_B T$ while the rest has $\epsilon_i^{ext} \leq 7\ k_B T$. We observed that the results shown in this paper are virtually insensitive to the choice of the $Z$ cutoff between 0.05 and 0.5 (see S1 Fig), therefore, in DEER-PREdict the default $Z$ cutoff can be conveniently replaced by a user-provided value.

## Predicting the DEER signal from structural ensembles

Electron-electron distance distributions extracted from DEER experiments, e.g. using the DeerLab package [36], have previously routinely been compared with distributions predicted using the RLA implemented in the Matlab-based MMM modeling toolbox (http://www.epr.ethz.ch/software) [13]. Since MMM intrinsically operates on single structures, we and others had to resort to wrapper scripts to compute distance distributions of large ensembles, such as MD trajectories [3, 25, 37]. With the program presented herein, we provide a tool to conveniently predict DEER distance distributions from large conformational ensembles, which can be easily integrated in reweighting schemes such as the Bayesian/maximum entropy procedure [1, 14, 38, 39].

For each trajectory frame or conformation of a given ensemble, the rotamers from the library are placed at the spin-labeled position (Fig 1A) and the distances between all pair combinations of N-O paramagnetic centers are calculated. The resulting matrix of pair-wise distances is then used to compute the distance distribution weighted by the combined probability of each probe conformation, $p_i \times p_j$, with $p_i$ and $p_j$ being the conformation probabilities of rotamers $i$ and $j$. After averaging over all the frames, a low-pass filter is applied to the distance distribution for noise reduction [40],

$$P(r) = \mathcal{F}\left\{\mathcal{F}^{-1}[P(r)] \times \mathcal{F}^{-1}\left[\exp\left(-\frac{r^2}{2\sigma^2}\right)\right]\right\} \tag{2}$$

where $\mathcal{F}$ and $\mathcal{F}^{-1}$ are the Fourier transform and inverse Fourier transform operators, respectively, whereas $\sigma$ is the standard deviation of the low-pass filter. The resulting $P(r)$ is a smooth curve even for the analysis of a single protein conformation (Fig 1B). The standard deviation of the low-pass filter can readily be provided by the user through the option *filter_stdev* of the *run* function in the *DEERpredict* class, overriding the default value of 0.5 Å. The average over

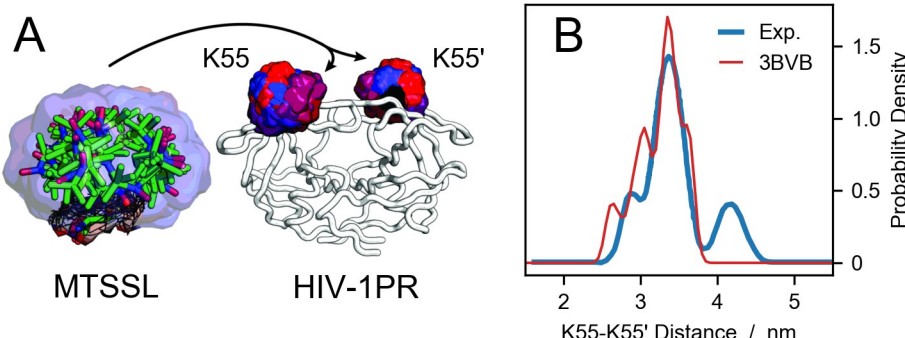

**Fig 1. Probe placement scheme and comparison to DEER data.** (A) A pool of 46 conformations of the MTSSL probe from the rotamer library are aligned to the backbone of residues K55 and K55' of HIV-1 protease. The color code represent the Boltzmann weights of each rotamer, increasing from blue to red. (B) Electron-electron distance distribution for HIV-1 protease spin labeled at residues K55 and K55'. The blue line is the experimental data from Torbeev *et al.* [44] whereas the red line is the prediction using DEER-PREdict and a crystal structure of HIV-1 protease (PDB code 3BVB).

the trajectory frames can be weighted by a user-specified list of weights e.g. to remove the bias from enhanced sampling simulations.

The dipolar modulation signal can be back-calculated from the distance distribution, $P(r)$, via the following integral [41]

$$S(t) = \int_0^\infty dr\, P(r) K(r, t). \tag{3}$$

$K(r, t)$ is the DEER kernel

$$K(r, t) = \sqrt{\frac{\pi}{6\omega t}} \left[ \cos(\omega)\, \mathrm{Fr\,C}\left(\sqrt{\frac{6\omega t}{\pi}}\right) + \sin(Dt)\, \mathrm{Fr\,S}\left(\sqrt{\frac{6\omega t}{\pi}}\right) \right] \tag{4}$$

where FrC and FrS are Fresnel cosine and sine functions, and $\omega$ is the dipolar frequency

$$\omega = \frac{\mu_0}{4\pi\hbar} \frac{\mu_B^2 g^2}{r^3} \tag{5}$$

where $\mu_0$ is the permeability of free space, $\mu_B$ is the Bohr magneton and $g$ is the electron g-factor. The ranges of inter-probe distance and time are $[0, r_{max}]$ and $[t_{min}, t_{max}]$ with increments $dr = 0.05$ nm and $dt$, respectively. The default values $r_{max} = 12$ nm, $t_{min} = 0.01$ $\mu$s, $t_{max} = 5.5$ $\mu$s and $dt = 0.01$ $\mu$s can be overridden by the user. Following the correction of the experimental DEER time trace for the intermolecular background [36, 42], the resulting form factor can directly be compared with

$$V(t) = 1 + \lambda[S(t) - 1] \tag{6}$$

where $0.02 \leq \lambda \leq 0.5$ is the modulation depth of the experimental signal [43], quantifying the efficiency of the DEER pump pulse [8].

## Prediction of PRE rates and intensity ratios

In analogy to the calculations of electron-electron distances to predict DEER distributions, we extended the use of the RLA to electron-proton separations to improve the accuracy of PRE predictions. We focus here is on PRE NMR experiments that probe the increase in transverse relaxation rates of any backbone proton due to the dipolar interaction with the unpaired

electron of the paramagnetic probe:

$$R_2^{ox} = R_2^{red} + \Gamma_2 \tag{7}$$

where $R_2^{ox}$ and $R_2^{red}$ are the transverse relaxation rates in the presence of the spin label in the oxidized or reduced (diamagnetic) state, respectively. We note that it is also possible to measure PREs on other atoms and to probe longitudinal relaxation enhancement, and it would be possible to include such measurements in future versions of DEER-PREdict.

A description of the enhancement of the transverse relaxation due to dipole-dipole interactions in paramagnetic solutions was first proposed by Solomon and Bloembergen [45, 46]

$$\Gamma_2 = \frac{1}{15}\left(\frac{\mu_0}{4\pi}\right)^2 \gamma_I^2 g^2 \mu_B^2 s_e(s_e + 1)[4J(0) + 3J(\omega_I)], \tag{8}$$

where $\gamma_I$ and $\omega_I$ are the gyromagnetic ratio and the Larmor frequency of the proton, respectively, whereas $s_e$ is the electron spin quantum number, equal to 1/2 for nitroxide probe systems. The spectral density function $J(\omega_I)$ can be described using a model-free formalism [47–50], which takes into account the overall molecular tumbling in the external magnetic field as well as the internal motion of the spin label:

$$J(\omega_I) = \langle r^{-6}\rangle\left[\frac{S^2\tau_c}{1 + \omega_I^2\tau_c^2} + \frac{(1 - S^2)\tau_t}{1 + \omega_I^2\tau_t^2}\right] \tag{9}$$

where

$$\tau_c = \left(\frac{1}{\tau_r} + \frac{1}{\tau_s}\right)^{-1} \tag{10}$$

and

$$\tau_t = \left(\frac{1}{\tau_r} + \frac{1}{\tau_s} + \frac{1}{\tau_i}\right)^{-1}. \tag{11}$$

$\tau_r$ is the rotational correlation time of the protein, $\tau_s$ is the effective electron correlation rate and $\tau_i$ is the correlation time of the internal motion (effective correlation time of the spin label). For MTSSL probes, $\tau_s \gg \tau_r$ and $\tau_c \approx \tau_r$ [51]. The value of $\tau_c$ depends on protein size and structure and is generally of the order of 1–10 ns [27, 52–55]. For $\tau_i$, values between 100 to 500 ps can be assumed, based on e.g. $^{15}$N spin relaxation rates and MD simulations [56, 57]. In general, $\tau_c$ and $\tau_i$ can be specified as user input in DEER-PREdict.

For the generalized order parameter, $S$, we use the factorization into contributions from radial and angular internal motions introduced by Brüschweiler *et al.* [49], $S^2 = S_{radial}^2 S_{angular}^2$. The expressions for $S_{radial}^2$ and $S_{angular}^2$ were derived from a jump model that treats the $N$ conformers of the rotamer library as $N$ discrete states with equal probabilities (1/$N$) [50]. In reality, the various dihedral angles of the spin label have different free energy barriers, resulting in residence times between jumps ranging from less than 1 to several ns [17].

$$S_{radial}^2 = \frac{\langle r^{-3}\rangle^2}{\langle r^{-6}\rangle} \tag{12}$$

where $r$ is the proton-electron distance and the brackets denote averages over the conformers

weighted by the respective Boltzmann weights, $p_i$, i.e. $\langle r^{-3} \rangle = \sum_i^N r_i^{-3} p_i$ and $\langle r^{-6} \rangle = \sum_i^N r_i^{-6} p_i$.

$$S_{angular}^2 = \left\langle \frac{3}{2} \cos^2 \Omega - \frac{1}{2} \right\rangle = \sum_{i,j}^N \left[ \frac{3}{2} \left( \frac{\mathbf{r}_i \cdot \mathbf{r}_j}{r_i r_j} \right)^2 - \frac{1}{2} \right] p_i p_j \qquad (13)$$

where $\Omega$ is the angle between the vectors $\mathbf{r_i}$ and $\mathbf{r_j}$, connecting a backbone proton with the *i*th and *j*th rotamer states, respectively. The relaxation enhancement rate for a single protein structure is calculated using Eq 8, and assuming that the motion of the paramagnetic label is much faster than the protein conformational changes, the ensemble average is estimated as

$$\langle \Gamma_2 \rangle = \sum_k^M w_k \Gamma_{2,k}, \qquad (14)$$

where *M* is the number of configurations or frames of the simulation trajectory. In the case of unbiased simulations, the statistical weights, $w_l$, are simply $1/M$. Optionally, a list of weights can be provided by the user, e.g. to reweight a biased MD simulation [58, 59] or to incorporate the prediction of the PRE rates into a Bayesian/maximum entropy reweighting scheme [1].

For samples with particularly high PRE rates it can be infeasible to obtain $\Gamma_2$ from multiple time-point measurements [60]. In such and other cases, the PRE is sometimes probed indirectly from the ratio of the peak intensities in $^1$H,$^{15}$N-HSQC spectra of the spin-labeled protein in the oxidized and reduced state. Assuming that the intensity of the proton magnetization decays exponentially—by transverse relaxation only—during the total INEPT time of the HSQC measurement [61], $t_d$, the intensity ratio is estimated as

$$\frac{I_{para}}{I_{dia}} = \frac{R_2^{red} \exp \left( -\Gamma_2 t_d \right)}{R_2^{red} + \Gamma_2}. \qquad (15)$$

## Requirements and installation

The main requirements are Python 3.6–3.8 and MDAnalysis 1.0 [30, 62]. In an environment with Python 3.6–3.8, DEER-PREdict can readily be installed through the package manager PIP by executing

```
1 pip install DEERPREdict
```

## Package stability

Tests reproducing DEER and PRE data for the protein systems studied in this article, as well as for a nanodisc [29], are performed automatically using Travis CI (travis-ci.com/github/KULL-Centre/DEERpredict) every time the code is modified on the GitHub repository. The same tests can also be run locally using the test running tool pytest.

## Results

In the following, we present applications of our tool to the prediction of DEER distance distributions and PRE intensity ratios of three folded proteins.

The code snippets reported in this section pertain to DEER-PREdict version 0.1.7. A Jupyter Notebook to reproduce the results shown below (*article.ipynb*) can be found in the *tests/data* folder on the GitHub repository. Up-to-date documentation is available at deerpredict.readthedocs.io.

## Case study 1: DEER data for HIV-1 protease

HIV-1 protease (HIV-1PR) is a homodimeric aspartic hydrolase involved in the cleavage of the gag-pol polyprotein complex. The inhibition of this process affects the life cycle of the HIV-1 virus, rendering it noninfectious [63]. The HIV-1PR monomer is composed of 99 residues and presents a structurally stable core region (residues 1-43 and 58-99) and a dynamic region characterized by a $\beta$-hairpin turn, called the flap (residues 44-57). The active site is located at the interstice between the core regions of the two monomers, in proximity to the catalytic D25 residues. This cavity is closed off by the dynamic flap regions, which are considered to act as a gate controlling the access to the active site. The dynamics of the flap regions are of utmost importance for the development of inhibitors, and have been extensively studied, both experimentally and *in silico* [44, 64–69]. Based on the relative position of the flaps, three main conformational states have been proposed. In X-ray crystallography, the closed state is typically observed for the ligand-bound enzyme (e.g. PDB codes 3BVB [70] and 2BPX [71]), the semi-open state is predominant for the apo form (e.g. PDB code 1HHP [72]) whereas the wide-open state has been observed for variants (e.g. PDB codes 1TW7 [73] and 1RPI [74]) [69]. In DEER measurements, these conformational states can be resolved by spin-labeling sites K55 and K55' (see S1 Text and S2 Fig).

To assess the predictive ability of DEER-PREdict, we generated conformational ensembles of the HIV-1PR homodimer via two different approaches: (a) a single 500-ns unbiased MD simulation, and (b) four independent 125-ns MD simulations restrained with experimental residual dipolar couplings (RDC) data [58, 75] from Roche *et al.* [65, 66] (see S1 Text for methodological details). The initial configuration of our simulations is the X-ray crystal structure of the active-site mutant D25N bound to the inhibitor Darunavir (PDB code 3BVB).

Fig 2 presents a comparison of experimental DEER distance distributions and echo intensity curves with predictions from simulation trajectories of 1,000 frames sampled every 0.5 ns. The echo intensity curves are calculated using Eq 6, where the λ is estimated to 0.0922 by fitting the experimental dipolar evolution function to the corresponding curve derived from the experimental $P(r)$ via Eq 3. For a single trajectory, the analysis is performed in 13 s on a 1.7 GHz processor by running the following code:

```
1 import MDAnalysis
2 from DEERPREdict.DEER import DEERpredict
3 u = MDAnalysis.Universe('conf.pdb','traj.xtc')
4 DEER = DEERpredict(u,residues =[55, 55],chains=['A','B'],
temperature = 298)
5 DEER.run()
```

The third line generates the MDAnalysis Universe object from an XTC trajectory and a PDB topology. The fourth line initializes the DEERpredict object with the spin-labeled residue numbers and the respective chain IDs. The fifth line runs the calculations and saves per-frame and ensemble-averaged data to *res-55-55.hdf5* and *res-55-55.dat*, respectively, as well as the steric partition functions of sites K55 and K55' to the file *res-Z-55-55.dat*.

In the experimental distance distribution, the main peak at ∼3.3 nm corresponds to the closed state whereas the second peak between 4 and 5 nm is characteristic of the wide-open state. The shoulder peak at ∼2.8 nm has been identified as an open-like state known as the curled/tucked conformation [9, 76, 77]. The results of our unbiased and restrained simulations are in substantial agreement with the findings of Roche *et al.* [65, 66], indicating that the flaps of the inhibitor-free HIV-1PR are predominantly in closed conformation. Compared to the distance distribution calculated from the starting configuration of PDB code 3BVB (see Fig 1), predictions based on MD trajectories more accurately reproduce the shape of the shoulder and the main peak of the experimental $P(r)$. Moreover, using the RDC data as restraints leads to a

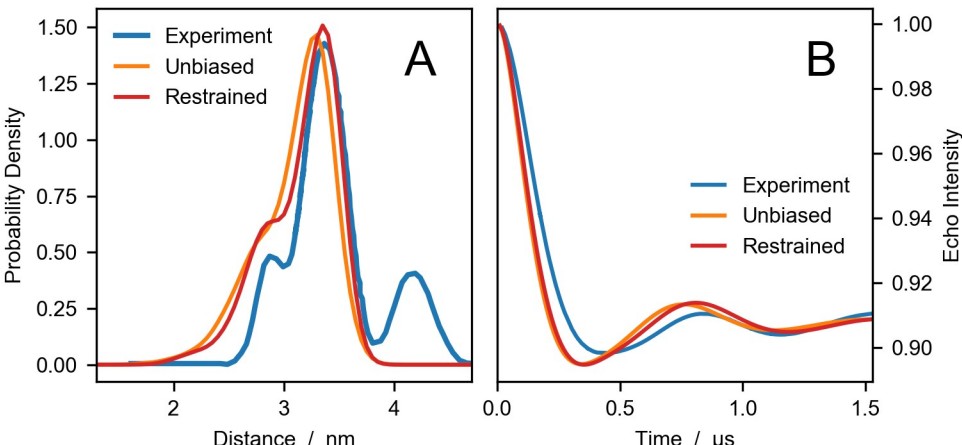

**Fig 2. Comparing experiments and simulations for HIV1-PR.** DEER distance distributions (A) and echo intensity curves (B) obtained by Torbeev *et al.* [44] from DEER experiments (blue), and calculated using DEER-PREdict from unbiased (orange) and RDC ensemble-biased MD simulations (red).

significant improvement in the agreement between simulations and experiments, with the RMSD decreasing from 0.07 for the unbiased to 0.03 for the RDC ensemble-biased simulations. However, in the simulations we do not observe the wide-open state. This discrepancy could be due to insufficient sampling or could be attributed to the difference in sequence between the simulated protein and the experimental construct.

## Case study 2: DEER data for T4 lysozyme

Lysozyme from the T4 bacteriophage (T4L) has long been used as a model system in the study of protein structure and dynamics [78–83]. Here, we focus on the L99A and the triple L99A-G113A-R119P mutants which are structurally similar and mainly differ in the relative populations of their major conformational states. The L99A variant presents a 150 Å$^3$ hydrophobic pocket capable of binding hydrophobic ligands and has been thoroughly studied to further our understanding of the dynamics and selectivity of the binding pocket [78, 84]. The L99A variant occupies two distinct conformational states: the ground state (G) and the transient excited state (E), amounting for 97% and 3% of the population, respectively. The large-scale motions converting the G into the E state occur on the millisecond time scale and result in the occlusion of the cavity, which is occupied by the side chain of F114 in the E state [82]. The additional G113A and R119P mutations in the triple-mutant variant interconvert the populations of the conformational states to 4% for the G state and 96% for the E state [82]—note that, here and in the following, we refer to the G and E states based on their structural similarity to the L99A variant rather than on their relative populations. These conformational equilibria have been studied by DEER for various pairs of spin-labeled sites, which effectively resolve the G and E states as separate peaks of the $P(r)$ [83].

Here, we compare DEER distance distributions calculated with DEER-PREdict for two pairs of probe positions (D89C–T109C and T109C–N140C) with the corresponding experimental data by Lerch *et al.* [83]. First, we calculate the $P(r)$ of the single states using PDB code 3DMV for the G states and PDB codes 2LCB and 2LC9 for the E states of single and triple mutants, respectively. Second, the $P(r)$'s are linearly combined based on the experimentally derived ratios of G and E populations (97:3 for L99A and 4:96 for L99A-G113A-R119P) [82]. Additionally, we predict DEER distance distributions from previously reported metadynamics

MD simulations of L99A and L99A-G113A-R119P [80] (see S1 Text for methodological details). In these calculations, the average over the trajectory is weighted by $\exp(F_{bias}/k_BT)$, where $F_{bias}$ is the final static bias for each frame and $k_BT$ is the thermal energy. The analysis of a trajectory of 6,670 frames is performed in 3 min on a 1.7 GHz processor executing the following lines of code:

```
1 import MDAnalysis
2 from DEERPREdict.DEER import DEERpredict
3 import numpy as np
4 u = MDAnalysis.Universe('conf.pdb','traj.xtc')
5 for residues in [[89, 109],[109, 140]]:
6   DEER = DEERpredict(u,residues = residues,temperature = 298,
z_cutoff = 0.1)
7   DEER.run(weights = np.exp(Fbias/(0.298*8.3145)))
```

In line six, we specify the positions of the spin-labels, the temperature at which the metadynamics simulations were performed and a non-default value for the $Z$ cutoff. In line seven, we provide the weights of each trajectory frame, generated from the array of $F_{bias}$ values.

Fig 3 shows a comparison between the experimental distance distributions obtained by Lerch *et al.* [83] and our predictions. In general, the calculated distributions fall within the experimental ranges of inter-probe distances and are particularly accurate for the D89C–T109C spin-labeled pair in metadynamics simulations. The sharper shape of the experimental $P(r)$'s, relative to the calculated distributions, could be due to the cryogenic temperatures at which DEER experiments are conducted, whereas simulations were performed at room temperature. For the T109C–N140C spin-labeled pair of the triple variant, the discrepancy between predicted and calculated $P(r)$'s might be explained by considering that distances shorter than 1.5 nm fall below the range probed by DEER experiments. On the other hand, the inaccurate predictions of the T109C–N140C $P(r)$ for the single (L99A) variant is greater than expected. Such discrepancies may be due both to errors in the protein structure or in the DEER-calculations. While our results cannot distinguish between these scenarios, we follow previous work [14] by examining whether the discrepancies can can be attributed to the error on the Boltzmann probabilities of the rotamer states, $p_i^{int}$. We thus use a Bayesian/maximum entropy (BME) procedure to show that a small change in the original rotamer weights

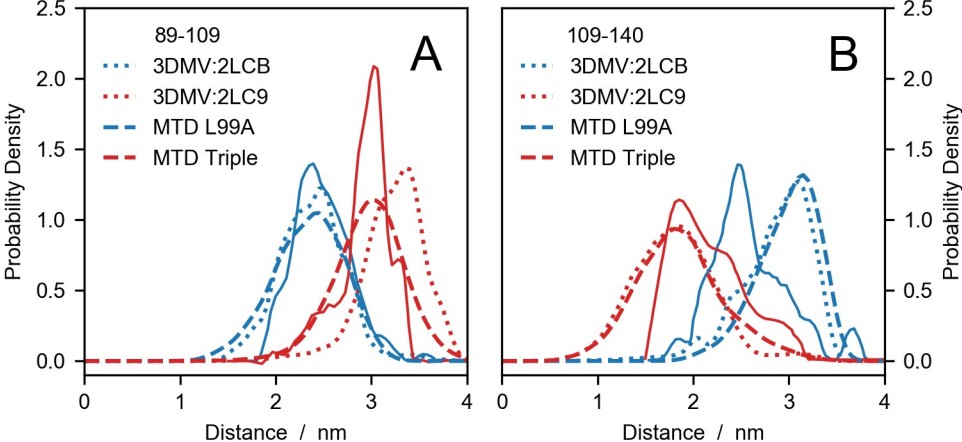

**Fig 3. Comparing experiments with simulations and structures of T4 lysozyme variants.** DEER distance distributions for probe positions (A) D89C–T109C and (B) T109C–N140C of the single (blue) and the triple variant (red). Solid lines are the experimental data by Lerch *et al.* [83], dotted lines are calculated from PDB codes and dashed lines are predictions from metadynamics (MTD) simulations by Wang and coworkers [80].

can lead to a substantial improvement of the agreement with the experimental data (see S1 Text and S3 Fig).

## Case study 3: PRE data for Acyl-CoA-binding protein

The RLA is well known in the EPR community and generally favored over e.g. a C$\alpha$-based approach as discussed elsewhere [3, 13, 26]. In the presented software, we apply the same improved modeling of the probe flexibility also to the prediction of PRE rates and intensity ratios.

Our test data is the PRE data for the bovine Acyl-coenzyme A Binding Protein (ACBP) reported by Teilum *et al.* [53]. In this study the structural behavior of ACBP under native and mildly-denaturing conditions was investigated via the SDSL of five positions in the amino acid sequence: T17C, V36C, M46C, S65C and I86C. Here, we focus on the native state of ACBP for which an NMR structure comprising 20 conformers has been refined from residual dipolar couplings (RDC) and deposited in the Protein Data Bank (PDB code 1NTI). Fig 4 shows a comparison between the experimental data and the intensity ratios calculated from the $\Gamma_2$ values averaged over the 20 conformations of the PDB entry. A good overall agreement is achieved across the different probe positions. Notably, using the RDC-refined structure, we reproduce most of the structural features observed in the PRE experiments, including the proximity of residues 24, 27, 31 and 34 to the spin-labeled residue 86, which is consistent with a helix-turn-helix motif. The predicted intensity ratios are generated in 1.5 s on a 1.7 GHz processor executing the following code:

```
1 import MDAnalysis
2 from DEERPREdict.PRE import PREpredict
3 u = MDAnalysis.Universe('1nti.pdb')
4 for res in [17, 36, 46, 65, 86]:
5   PRE = PREpredict(u,res,temperature = 298,atom_selection='H')
6   PRE.run(tau_c = 2e-09,tau_t = 2*1e-10,delay = 1e-2,r_2 = 12.6,
wh = 750)
```

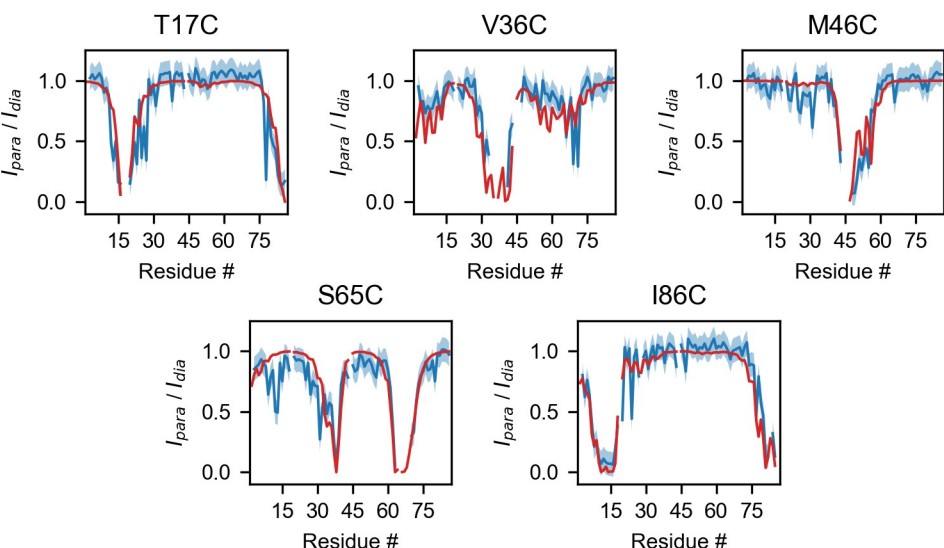

**Fig 4. Calculated and experimental PRE HSQC intensity ratios for the T17C, V36C, M46C, S65C and I86C mutants of ACBP.** Blue lines represent the experimental data [53], with the associated ±0.1 error shown by the blue shaded areas. Red lines represent intensity ratios calculated from PDB code 1NTI with $\tau_c$ = 2 ns, $\tau_t$ = 0.2 ns, $t_d$ = 10 ms, $R_2$ = 12.6 s$^{-1}$.

At line three, we load PDB code 1NTI as an MDAnalysis Universe object. We then use a for loop to calculate the PRE data from the distances between amide protons and the spin-label N-O group at five different positions along the amino acid sequence. In the last line we specify $\tau_c$ = 2 ns, $\tau_t$ = 0.2 ns, $t_d$ = 10 ms, $R_2$ = 12.6 s$^{-1}$ and $\omega_I$ = $2\pi \times 750$ MHz. Per-frame and ensemble-averaged PRE data are automatically saved to files named *res-\*.pkl* and *res-\*.dat*, respectively, whereas per-frame steric partition functions are saved to *res-Z-\*.dat*.

As detailed in S1 Text and S4 Fig, the steric partition functions provided by DEER-PREdict can be used to predict whether a position in the sequence is likely to accommodate the paramagnetic probe within the wild-type structure. Besides aiding the interpretation of experimental data, this feature can be instrumental to designing and enhancing the success-rate of time- and labor-intensive SDSL experiments.

As previously discussed, the explicit treatment of the paramagnetic probe may be crucial for the accurate back-calculation of DEER data, and even more so for PRE predictions, due to the $\langle r^{-6} \rangle$-dependence of the PRE. A common way to restrain MD simulations or to back-calculate PRE experimental data without explicitly simulating the paramagnetic probe is to approximate the electron location to the position of the C$\beta$ atom of the spin-labeled residue [85]. The advantage of this approach is that (a) multiple labeling sites can be analyzed in a single simulation and (b) the explicit atom is present in the simulation making the calculation of PREs straightforward. C$\beta$-based calculations may, however, be prone to over- or underestimating electron-proton distances by several Å, thereby introducing a systematic error. The impact of the C$\beta$-approximation on the accuracy of PRE predictions is illustrated in S5 and S6 Figs for the case of ACBP (see also S1 Text).

## Conclusion

We have introduced an open-source software program with a fast implementation of the RLA in tandem with protein ensemble averaging, for the calculation of DEER and PRE data. Using three examples, we have highlighted the capabilities of our implementation: (a) the extension of the RLA for DEER data from a protein ensemble and (b) the calculation of PRE rates and intensity ratios with the same approach.

The structural interpretation of DEER and PRE measurements requires an accurate treatment of the structure and conformational heterogeneity of the spin labels. In the presented software, this is achieved using the RLA and, in the case of the PRE, a model-free approach to describe the dynamics. Relative to simulations of the explicitly spin-labeled mutants, the RLA presents the particular advantage of enabling the prediction for multiple SDSL experiments from a single simulation of the wild type sequence.

## Availability and future directions

The software is implemented using the popular trajectory analysis package MDAnalysis, version 1.0 [30] and is available on GitHub at github.com/KULL-Centre/DEERpredict. DEER-PREdict is also distributed as a PyPI package (pypi.org/project/DEERPREdict) and archived on Zenodo (DOI: 10.5281/zenodo.3968394). DEER-PREdict and MDAnalysis are published and distributed under GPL licenses, version 3 and 2, respectively.

DEER-PREdict has a general framework and can be readily extended to encompass non-protein biomolecules as well as additional rotamer libraries of paramagnetic groups. Moreover, the software can be augmented with a module to predict Förster resonance energy transfer data, combining the insertion routines already implemented for MTSSL probes with rotamer libraries for fluorescent dyes.

## Supporting information

**S1 Text. Supporting information for "DEER-PREdict: Software for efficient calculation of spin-labeling EPR and NMR data from conformational ensembles".**
(PDF)

**S1 Fig. Influence of $Z$ cutoff on predicted DEER and PRE NMR data.** (A) DEER distance distributions calculated from RDC ensemble-biased MD simulations of HIV-1PR. (B) Predicted intensity ratios for ACBP spin-labeled at position 86 obtained from PDB code 1NTI with $\tau_c$ = 2 ns, $\tau_t$ = 0.2 ns, $t_d$ = 10 ms, $R_2$ = 12.6 s$^{-1}$. DEER and PRE predictions are performed using three different cutoff values of the steric partition function, $Z$, namely 0.05 (blue lines), 0.5 (orange lines) and 0.8 (red lines).
(TIF)

**S2 Fig. Comparison of DEER data from Torbeev *et al.* [44] with X-ray crystal structures deposited in the Protein Data Bank.** DEER distance distributions (A) and echo intensity curves (B) obtained by Torbeev *et al.* [44] from DEER experiments (blue), and calculated using X-ray crystal structures representative of closed (PDB code 2BPX, orange), semi-open (PDB code 1HHP, green) and wide-open (PDB code 1TW7, red) HIV-1PR conformations.
(TIF)

**S3 Fig. Optimization of rotamer weights using a Bayesian/maximum entropy procedure.** (A) $\chi^2$ vs $\varphi_{eff}$ for various values of the confidence parameter, $\theta$. (B) Distance distributions calculated from PDB codes 3DMV and 2LCB, using optimized weights obtained for various $\theta$ values. (C) Original [24] and modified weights of the MTSSL 175 K rotamer library after BME reweighting with $\theta$ = 4. DEER distance distributions for probe positions (D) D89C–T109C and (E) T109C–N140C of the single (blue) and the triple variant (red). Solid lines are the experimental data by Lerch *et al.* [83]; dotted and dashed lines are from PDB codes 3DMV, 2LC9 and 2LCB using the original and the BME-reweighted ($\theta$ = 4) MTSSL 175 K rotamer library.
(TIF)

**S4 Fig. Steric partition function quantifying the fitness of the rotamers at the spin-labeled site.** Steric partition function calculated from rotamer-protein van der Waals interactions for five spin-labeled mutants of ACBP. The horizontal dashed line indicates the cutoff used in the criterion for discarding protein conformations where the placement of the rotamer is characterized by steric clashes with the surrounding residues.
(TIF)

**S5 Fig. Comparison between RLA and C$\beta$-based PRE predictions.** PRE intensity ratios for ACBP spin labeled at position 65 calculated for (A) $\tau_c$ = 2 ns and (B) $\tau_c$ = 0.5 ns. Blue lines represent the experimental data [53], with the associated ±0.1 error shown by the blue shaded areas. Orange and red lines represent C$\beta$-based and RLA-based predictions, respectively.
(TIF)

**S6 Fig. Dependence on $\tau_c$ of the RMSD between experimental and predicted PRE ratios of ACBP: Comparison of optimal $\tau_c$ values for RLA vs. C$\beta$-based approach.** Red and blue lines are obtained using the RLA and approximating the electron location with the position of the C$\beta$ atom, respectively. Solid and dashed lines represent the RMSD values calculated from all the data points and from intensity ratios in the dynamic range $0.1 < I_{para} / I_{dia} < 0.9$.
(TIF)

## Acknowledgments

We thank Robert Best for help with RDC-restrained simulations as well as work on extending DEER-PREdict to use for prediction of FRET experiments.

## Author Contributions

**Conceptualization:** Giulio Tesei, João M. Martins, Micha B. A. Kunze, Kresten Lindorff-Larsen.

**Data curation:** Giulio Tesei, João M. Martins, Micha B. A. Kunze, Yong Wang.

**Formal analysis:** Giulio Tesei, João M. Martins.

**Funding acquisition:** Ramon Crehuet, Kresten Lindorff-Larsen.

**Investigation:** Giulio Tesei, João M. Martins, Micha B. A. Kunze, Ramon Crehuet, Kresten Lindorff-Larsen.

**Methodology:** Giulio Tesei, João M. Martins, Micha B. A. Kunze, Ramon Crehuet, Kresten Lindorff-Larsen.

**Project administration:** Micha B. A. Kunze, Kresten Lindorff-Larsen.

**Resources:** Kresten Lindorff-Larsen.

**Software:** Giulio Tesei, João M. Martins, Micha B. A. Kunze, Ramon Crehuet.

**Supervision:** Micha B. A. Kunze, Kresten Lindorff-Larsen.

**Validation:** Giulio Tesei, João M. Martins, Micha B. A. Kunze, Ramon Crehuet.

**Visualization:** Giulio Tesei, João M. Martins.

**Writing – original draft:** Giulio Tesei, João M. Martins, Micha B. A. Kunze, Kresten Lindorff-Larsen.

**Writing – review & editing:** Giulio Tesei, João M. Martins, Micha B. A. Kunze, Yong Wang, Ramon Crehuet, Kresten Lindorff-Larsen.

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
