## [Decision Letter · Decision Letter 0]

21 Oct 2020

Dear Dr. Lindorff-Larsen,

Thank you very much for submitting your manuscript "DEER-PREdict: Software for Efficient Calculation of Spin-Labeling EPR and NMR Data from Conformational Ensemble" for consideration at PLOS Computational Biology. As with all papers reviewed by the journal, your manuscript was reviewed by members of the editorial board and by several independent reviewers. The reviewers appreciated the attention to an important topic. Based on the reviews, we are likely to accept this manuscript for publication, providing that you modify the manuscript according to the review recommendations.

Sincerely,

Dina Schneidman

Software Editor

PLOS Computational Biology

[LINK]

Reviewer's Responses to Questions

**Comments to the Authors:**

Reviewer #1: This manuscript reports on a very useful software for predicting distance distributions for spin label pairs as well as experimental DEER/PELDOR data and paramagnetic relaxation enhancement (PRE) data from MD trajectories. The software is implemented in a user-friendly way and is computationally efficient. Especially for PRE it enables better predictions than are currently made by most practitioners in the field, as demonstrated in the Supporting Information. In general, the performance is nicely illustrated on the three application examples, although there is some unexplained discrepancy for one site pair and one mutant for T4 Lysozyme (see below). The manuscript is well written and concise. I recommend minor revision, taking into account the suggestions below.

Details:

1. In line 14, you quote a distance range from ~2 to ~6 nm. This is outdated information. A 2012 review (doi: 10.1146/annurev-physchem-032511-143716) already quoted 8 nm without protein deuteration and 11.5 nm with protein deuteration for soluble proteins. A 2016 paper (doi: 10.1002/anie.201609617) demonstrated 16 nm on a fully deuterated protein, admittedly in an exceptional case. In any case, the 6 nm limit has been surpassed in many application examples.

2. Line 73: It would be useful to specify what information must be included in a rotamer library.

3. Line 105: I believe that Ref. 34 does not relate to MMM.

4. Line 118: Please specify the low pass filter. It leads to some broadening of the distance distribution, which is fine, as the rotamer approach neglects some contributions to conformational distribution, but this broadening should not be excessive.

5. Below Eq (3): Please specify the kernel (minimum and maximum r, r increment, maximum t, t increment) or state that these values can be set by the user and provide recommended settings (especially for the increments).

6. The result of Eq. (5) cannot be directly compared to the experimental DEER trace. It describes the so-called form factor after correction for the intermolecular background. Either state this (and give a reference) or provide an explanation (and equations) for including this background. In general, it would be useful to refer to pre-processing that is necessary. Descriptions can be found in either Ref. 34 (a Python package amenable to pipelining with your software) or the older doi: 10.1007/BF03166213.

7. Line 144: “paramagnetic spin number” could be read as number of electron spins, but it is the electron spin quantum number.

8. Line 160: In reality, the jump rates between rotamers vary strongly (see Ref. 15). You might want to point out that this PRE model with a single correlation time is an approximation. It would also be useful to mention the time scale for such “jumps” implied by the results in Ref. 15.

9. Results for T4L: For site pair 89-109 (Fig. 3A) the agreement of metadynamics simulations with experiments is as good as one can expect. For site pair 109-140, the experimental result for the triple mutant (red in Fig. 3B) probably misses the shorter distances predicted by metadynamics because they fall outside the DEER range (you may want to comment on that). However, for mutant L99A the metadynamics simulation is clearly off, well beyond expected combined error of experiment and rotamer approach, whereas the prediction from PDB 2LCB fits experimental data almost perfectly. Either this is a freezing-related problem or a problem with the metadynamics simulation for this mutant. In any case, the discrepancy needs to be mentioned and discussed to some extent.

Typos:

Line 5: “spacial resolution” should read “spatial resolution”

Reviewer #2: The authors introduce DEER-PREdict, a software to calculate EPR and NMR data from computational ensembles of spin-label biomolecules. DEER-PREdict is an important methodological advance for employing data from EPR (DEER) and NMR (PRE) experiments in integrative modelling of biomolecules.

It is very useful that the DEER-PREdict provides implementations of the calculation of the DEER and PRE signals. Direct comparison to the experimental data are vital when assessing ensembles from computational modelling or trying to understand patterns in the experimental data.

The examples in the manuscript are highly relevant and help to illustrate the power of this approach. E.g., the agreement shown for the PRE data of ACBP (in Fig. 4) shows that very good agreement can obtained and that taking the spin labels into account captures features that otherwise cannot be reproduced (Fig. S4).

DEER-PREdict is easy to install and use. The software is nicely written and the code is straightforward to understand. Comments in the code further aid understanding. Importantly, automatic tests ensure that further modifications do not break the code.

DERR-PREdict will become a very useful tool for structural biologists and biophysicists.

**Have all data underlying the figures and results presented in the manuscript been provided?**

Reviewer #1: Yes

Reviewer #2: None

PLOS authors have the option to publish the peer review history of their article (what does this mean?). If published, this will include your full peer review and any attached files.

Reviewer #1: **Yes: **Gunnar Jeschke

Reviewer #2: No
---

## [Editor Report · Decision Letter 1]

19 Nov 2020

Dear Dr. Lindorff-Larsen,

We are pleased to inform you that your manuscript 'DEER-PREdict: Software for Efficient Calculation of Spin-Labeling EPR and NMR Data from Conformational Ensemble' has been provisionally accepted for publication in PLOS Computational Biology.

Best regards,

Dina Schneidman

Software Editor

PLOS Computational Biology

---

## [Editor Report · Acceptance letter]

18 Jan 2021

PCOMPBIOL-D-20-01460R1 

DEER-PREdict: Software for Efficient Calculation of Spin-Labeling EPR and NMR Data from Conformational Ensembles

Dear Dr Lindorff-Larsen,

I am pleased to inform you that your manuscript has been formally accepted for publication in PLOS Computational Biology. Your manuscript is now with our production department and you will be notified of the publication date in due course.

With kind regards,

Jutka Oroszlan
